# Role of SALL4 in HER2+ Breast Cancer Progression: Regulating PI3K/AKT Pathway

**DOI:** 10.3390/ijms232113292

**Published:** 2022-10-31

**Authors:** Birlipta Pattanayak, Ana Lameirinhas, Sandra Torres-Ruiz, Octavio Burgués, Ana Rovira, María Teresa Martínez, Marta Tapia, Sandra Zazo, Joan Albanell, Federico Rojo, Begoña Bermejo, Pilar Eroles

**Affiliations:** 1Biomedical Research Institute INCLIVA, 46010 Valencia, Spain; 2Department of Pathology, Hospital Clínico Universitario de Valencia, 46010 Valencia, Spain; 3Center for Biomedical Network Research on Cancer (CIBERONC), 28029 Madrid, Spain; 4Cancer Research Program, IMIM (Hospital del Mar Medical Research Institute), 08003 Barcelona, Spain; 5Department of Medical Oncology, Hospital Clínico Universitario de Valencia, 46010 Valencia, Spain; 6Department of Pathology, Fundación Jiménez Díaz, 28040 Madrid, Spain; 7Department of Medical Oncology, Hospital del Mar, 08003 Barcelona, Spain; 8Department of Physiology, Universidad de Valencia, 46010 Valencia, Spain; 9Department of Biotechnology, Universidad Politécnica de Valencia, 46022 Valencia, Spain

**Keywords:** SALL4, PI3K/AKT pathway, EMT, HER2+ breast cancer

## Abstract

Treatment for the HER2+ breast cancer subtype is still unsatisfactory, despite breakthroughs in research. The discovery of various new molecular mechanisms of transcription factors may help to make treatment regimens more effective. The transcription factor SALL4 has been related to aggressiveness and resistance therapy in cancer. Its molecular mechanisms and involvement in various signaling pathways are unknown in the HER2+ breast cancer subtype. In this study, we have evaluated the implication of SALL4 in the HER2+ subtype through its expression in patients’ samples and gain and loss of function in HER2+ cell lines. We found higher SALL4 expression in breast cancer tissues compared to healthy tissue. Interestingly, high SALL4 expression was associated with disease relapse and poor patient survival. In HER2+ cell lines, transient overexpression of SALL4 modulates PI3K/AKT signaling through regulating PTEN expression and BCL2, which increases cell survival and proliferation while reducing the efficacy of trastuzumab. SALL4 has also been observed to regulate the epithelial–mesenchymal transition and stemness features. SALL4 overexpression significantly reduced the epithelial markers E-cadherin, while it increased the mesenchymal markers β-catenin, vimentin and fibronectin. Furthermore, it has been also observed an increased expression of MYC, an essential transcription factor for regulating epithelial-mesenchymal transition and/or cancer stem cells. Our study demonstrates, for the first time, the importance of SALL4 in the HER2+ subtype and partial regulation of trastuzumab sensitivity. It provides a viable molecular mechanism-driven therapeutic strategy for an important subset of HER2-overexpressing patients whose malignancies are mediated by SALL4 expression.

## 1. Introduction

Breast cancer (BC) is one of the foremost cancers, affecting women worldwide [1]. BC has been revealed as a very heterogeneous disease with several intrinsic molecular subtypes. Among them, the human epidermal growth factor receptor 2 (HER2) positive subtype is one of the most vulnerable and susceptible subtypes for cancer progression and disease relapse [2]. Many studies have focused on a better understanding of molecular insight into HER2 signaling pathways that lead to aggressive disease and poor prognosis [3]. Tyrosine kinase receptor HER2 is also referred to as an orphan receptor because it lacks a particular binding ligand [4]. This receptor preferentially forms homo- or heterodimer with all other HER receptors and activates two signaling cascades downstream [4].

The phosphoinositide 3-kinase/ protein kinase B (PI3K/AKT) pathway is mostly implicated in cell proliferation and tumor growth [5]. It is well established that this pathway plays a vital role in providing nutrients, hormones and growth factors to cancer cells, facilitating their multiplication and tumor growth [6]. Upon activation of this pathway, phosphatase and tensin homologue deleted on chromosome 10 (PTEN) expressions tend to be reduced, which acts as an anti-tumor gene in different cancer types [7]. Several studies showed that mutation or inactivation of PTEN are often associated with HER2 amplification and, as a consequence, acquired resistance to different targeted therapies, such as trastuzumab, a monoclonal antibody used in the treatment of HER2+ BC subtype, can develop. Therefore, restoring PTEN expression or repairing its mutation has become an open challenge for scientists, which may lead to a promising therapeutic strategy for HER2+ cancer patients [8].

The human homologue of Drosophila spalt (sal) homeotic gene, SALL4, encodes a C_2_H_2_ zinc finger transcription factor (TF) [9]. This TF is mostly involved in embryonic development including organogenesis, neural development and limb formation. Any abnormal expression or mutation in this gene leads to the progression of various diseases and the development of different cancers [10]. In recent years, SALL4 has emerged as a potential biomarker in many cancers, including BC [11,12,13,14]. Researchers have observed that 86.1% of BC cases have shown high SALL4 expression, and it is regulating different tumor suppressor genes, including E-cadherin (CDH1) and PTEN [15]. It was also observed that SALL4 expression is associated with drug resistance to different chemotherapy agents in BC, including carboplatin (Car) [16], doxorubicin (Dox) [17] and cisplatin (Cis) [18]. Some studies have suggested that SALL4 silencing in Dox BC resistance cell lines reverts the resistance through cell cycle arrest and downregulation of membrane transporter ABCG2, more commonly referred to as BCRP (breast cancer resistance protein) [17] and nuclear receptor-binding protein 1 (NRBP1) [19]. Other studies in different tumor types have also provided insightful evidence regarding SALL4 involvement in drug resistance. As evidence, it has been shown that SALL4 is related to Car resistance in endometrial cancer by inducing epithelial-mesenchymal transition (EMT) and MYC expression, whereas reducing SALL4 expression enhances Car efficiency in endometrial cancer cells [20]. In addition, SALL4 has been reported to interact with the nucleosome remodeling and deacetylase (NuRD) complex and regulate the PI3K/AKT pathway [21].

In our study, we showed for the first time that SALL4 is associated with HER2+ BC progression through inducing proliferation, EMT, and maintaining stemness. Furthermore, we showed that SALL4 activated PI3K/AKT pathway, by controlling PTEN expression, which led to high cell proliferation and partially reduced the effect of trastuzumab. In conclusion, these findings indicate a novel mechanism of SALL4 in driving HER2+ BC development and could also be a potential prognostic marker for this BC subtype.

## 2. Results

### 2.1. SALL4 Expression in BC Patient’s Samples

Dysregulation of SALL4 gene is a known cause of cancer. In BC, particularly in the HER2+ BC subtype, its expression is less explored. To assess the RNA expression of this gene, RNA-seq data were retrieved from the public OncoDB database from the breast invasive carcinoma tissues and divided into two groups, tumor (*n* = 1135) vs. normal (*n* = 114) samples. We found that SALL4 expression was significantly lower in healthy tissues compared to tumor samples (*p*-value = 7.3 × 10^−114^) (Figure 1A). In order to compare SALL4 expression in different BC intrinsic subtypes, the TCGA database was used. Where we obtained 523 numbers of specimen details, others had missing SALL4 expression data, no follow-up data or missing clinical information. Among them, there were luminal B (*n* = 127), basal-like (*n* = 98), luminal A (*n* = 231), HER2+ (*n* = 58) and normal-like tumor samples (*n* = 9), which showed the highest expression of SALL4 (*p*-value = 0.0013) in HER2-enriched tissue samples followed by luminal A, luminal B and basal-like subtypes (Figure 1B). The low-level expression was observed in the normal-like group which was also taken as a reference to analyze the *p*-value. The analysis of 18 HER2+ BC specimens, 7 responders and 11 non-responders to trastuzumab treatment showed higher SALL4 (*p*-value = 0.0340) expression in non-responding patients compared to the responders (Appendix A). 

In order to evaluate SALL4 implication in BC patient survival, we analyzed the overall survival (OS) and distant metastasis-free survival (DMFS) in all types of BC (ALL) and, specifically, in the HER2+ BC subtype related to this gene expression. The results showed that more expression of SALL4 was significatively related to less OS (*p*-value = 0.0058) and reduced DMFS (*p*-value = 0.0011) for ALL BC (Figure 1C,D). Regarding HER2+ BC, we have evaluated the relationship between SALL4 expression and survival using three different HER2+ databases (PAM50, HER2 array and StGallen) of the Kaplan–Meier Plotter tool. High levels of SALL4 expression correlated significantly with low DFMS in all three databases (*p*-value =0.045, *p*-value =0.0026, *p*-value = 0.00076, respectively). However, the correlation between elevated SALL4 expression and OS was significant in HER2 array (*p*-value = 0.0023) and StGallen (*p*-value = 0.0016) databases analysis but not in PAM50 analysis, where it showed a trend without reaching significance. (Figure 1E,F and Appendix A). These results pointed out that SALL4 overexpression in BC is highly correlated with poor prognosis and suggest SALL4 as a potential predictor of metastases development and patient survival in BC. With regard specifically to the HER2+ BC subtype, the trend in the data suggests a similar effect.

### 2.2. Ectopically SALL4 Expression Enhances Proliferation and Hinders Partially the Trastuzumab Effect

To understand the implications of SALL4 in cell proliferation, the gene was ectopically expressed in two HER2+ BC cell lines (BT474 and SKBR3). The data showed a significant increase in proliferation when cells overexpressed the gene (*p*-value = 0.0023 for BT474, *p*-value = 0.0017 for SKBR3). Interestingly, cells with SALL4 overexpression decreased their trastuzumab response compared to cells with basal SALL4 levels (*p*-value = 0.0286 for BT474, *p*-value = 0.0003 for SKBR3) due to its higher proliferation ratio (Figure 2A–D).

Additionally, the silencing of SALL4 in the BT474R cell line was performed, where it showed a high basal expression at the gene and protein level (Appendix A). The SALL4 downregulation in this cell line showed a significant decrease in cell proliferation (*p*-value = 0.0006 for si#1 and *p*-value = 0.0020 for si#2) and, accordingly, a better response to trastuzumab treatment compared to transfection control (*p*-value = 0.0007 for si#1 and *p*-value = 0.0003 for si#2) (Appendix A). 

In summary, these findings indicate that SALL4 overexpression increases cell proliferation, which may be partially involved in drug resistance in HER2+ BC cells and helps cells escape from treatment. 

### 2.3. SALL4 Follows the PI3K/AKT Pathway for Cell Proliferation

One of the foremost impactful flagging pathways regulated by HER2 overexpression is the PI3K/AKT signaling pathway, which influences cell cycle progression and can repress apoptosis. Given the essential role of this pathway in HER2+ BC survival and proliferation, we hypothesized that SALL4 might play an important role in regulating cell proliferation and partially impeding trastuzumab resistance through the PI3K/AKT pathway.

The SALL4 gain of function in HER2+ BC cell lines (BT474 and SKBR3) was performed to evaluate its impact on PI3K/AKT pathway. It was found that SALL4 overexpression activates PI3K and AKT at tyrosine and serine residues, respectively, and represses PTEN expression. This also led to the activation of AKT downstream targets, such as BCL2 (Figure 2E,F). In contrast, in the acquired resistance HER2+ BC cell line (BT474R), the ectopically down-regulation of SALL4 decreased PI3K and AKT’s activation and upregulated PTEN expression, along with downregulation of BCL2 (Appendix A). These results suggested that SALL4 regulates the HER2+ pathway through the PI3K/AKT pathway, which leads to cell growth and tumor proliferation, favoring trastuzumab resistance, and may play a significant role in BC progression.

### 2.4. RBBp4 as a Bridge between SALL4 and PI3K/AKT

The retinoblastoma binding protein 4 (RBBp4) is an essential subunit of the NuRD complex, which plays a crucial role in the progression of different cancers [22]. It has been observed that the interaction between SALL4 and RBBp4 may recruit the NuRD complex to the PTEN promoter; as a consequence, PTEN repression and activation of PI3K/AKT pathway are produced, enhancing cancer cell proliferation [23]. 

We co-immunoprecipitated SALL4 and RBBp4 to demonstrate that these proteins directly interact with one another in our experimental model, thereby validating the physical interaction of SALL4-RBBp4 (Appendix A). To assess the potential role of RBBp4 in cancer of HER2+ BC progression due to interaction with SALL4 in HER2+ BC, we analyzed RBBp4 expression in BC patient samples from public databases. We found that the expression of RBBp4 is significantly lower in healthy tissues (*n* = 114) compared to BC (*n* = 1135) samples (*p*-value = 4.5 × 10^−13^) (Figure 3A) from oncoDB database. We also analyze the basal expression of RBBp4 in the different BC molecular subtypes. We could obtain 976 details of specimens from the TCGA database, among them, there were luminal B (*n* = 194), basal-like (*n* = 142), luminal A (*n* = 434), HER2+ (*n* = 87) and normal-like tumor sample (*n* = 119). Almost all molecular subtypes of BC showed high expression of RBBp4 except the normal-like group, taken as reference (Figure 3B). According to these results, we hypothesized that RBBp4 can be involved in the activation of the PI3K/AKT pathway together with SALL4. As a consequence, the repression of PTEN would take place and this would enhance the proliferation of HER2+ BC cell lines, helping the cells to escape from drugs. Therefore, the regulation of NuRD complex units’ expression was evaluated after performing the SALL4 gain and loss function. The SALL4 overexpression in BT474 and SKBR3 HER2+ BC cell lines showed a positive regulation of RBBp4, MTA1, MBD3, HDAC1 and HDAC2 protein expression (Figure 3C,D). The opposite effect was observed after SALL4 silencing in the resistant cell line (Appendix A), where RBBp4, MTA1, MBD3, HDAC1 and HDAC2 have been downregulated. Taken together, these data suggested that the interaction between SALL4 and RBBp4 (NuRD complex) inhibits PTEN expression and activates the PI3K/AKT pathway that helps cells to survive and to resist therapy.

### 2.5. SALL4 Promotes EMT and Stemness

Considering that SALL4 is a TF that has been reported to play an essential role in maintaining stem cell-like states and leading to EMT processes, we evaluate EMT and stemness markers in BT474 and SKBR3 cell lines that ectopically overexpressed SALL4. The SALL4 gain of function in HER2+ cell lines induced a significant reduction in the epithelial marker CDH1 expression, whereas the expression of the mesenchymal-like β-catenin (CTNNB1), vimentin (VIM) and fibronectin (FN1) have increased in both mRNA and protein level. Furthermore, it has been also observed that MYC, an essential TF for regulating EMT and/or cancer stem cells, showed increased expression (Figure 4A–D). Meanwhile, we observed the opposite effect in the resistance cell line when we silenced the SALL4 expression (Appendix A). As EMT and stemness are interlinked processes, we also evaluated the expression of some stemness markers and observed that OCT4, SOX2 and NANOG were upregulated in HER2+ BC cell lines that ectopically overexpressed SALL4 (Figure 4E–H). In contrast, those stemness markers were downregulated when silencing SALL4 in the trastuzumab-resistant cells (Appendix A). These results suggest that overexpressing of SALL4 promotes EMT and stemness, which is required for the cells to be more aggressive and able to resist treatments. 

## 3. Discussion

HER2+ BC is an aggressive disease that is more likely to reoccur than luminal A and luminal B BC subtypes [24]. Still, while some HER2+ BC patients can experience recurrence, recent advancements in anti-HER2 targeted therapies and long-term treatment approaches have decreased relapse rates. To improve treatment approaches, it is essential to explore the different molecular mechanisms that will guide to advance of targeted therapies and prevent disease recurrence more effectively. Among them, TFs play a critical role in controlling various direct mechanisms including chromosomal translocations, gene amplification or deletion, point mutations and alteration of expression, and indirectly through non-coding DNA mutations that affect TF binding [25]. Despite several innovative strategies to target TFs, they are still insufficient to completely eradicate the illness. [25,26,27]. Therefore, exploring new molecular mechanisms, which drive tumor aggressiveness and therapy resistance via different TFs is essential to advance the treatments. 

SALL4 is a TF that plays essential roles during embryonic development and has also been shown to be involved in various cancer-related processes such as cell proliferation, EMT, maintaining stemness, drug resistance, metastasis, invasion and migration [15,27,28,29]. However, the implication of SALL4 in BC is still unclear, once the majority of the studies are focused on its role in the basal-like subtype. As previously mentioned, we found a higher SALL4 expression in BC samples compared with normal breast tissue [30]. Moreover, we showed that SALL4 expression levels were also related to worse outcomes in BC, particularly in the HER2+ BC subtype. Furthermore, elevated SALL4 levels were significantly associated with clinical trastuzumab resistance. In this study, we explored SALL4 implications in the HER2+ BC subtype. For that, SALL4’s gain and loss of function experiments were performed in parental HER2+ and acquired trastuzumab-resistant cell lines, respectively. Our results showed that SALL4 downregulation led to a decrease in proliferation, whereas an increased proliferation was observed when SALL4 was over-expressed. Contrarily, SALL4 downregulation resulted in a partial restoration of trastuzumab sensitivity, which is consistent with the function of SALL4 expression on controlling cell growth. These results are also in agreement with the theory of Kobayashi et al., where NANOG and SALL4 are described as vital factors for maintaining an undifferentiated state and cell proliferation, respectively [31]. 

Further, to clarify the mechanism under SALL4´s influence on cell proliferation and trastuzumab effectiveness, we decided to explore the PI3K/AKT pathway, as this pathway is one of the most studied in HER2+ BC. Indeed, one of the most common mechanisms behind trastuzumab resistance disease is the mutational activation of this pathway and changes in the HER2 molecule itself. In particular, mutational activation in PIK3CA, loss of PTEN, increased expression of p95-HER2 and loss of expression of HER2 have been proposed to contribute to trastuzumab resistance [32]. Previous studies have already suggested that silencing SALL4 in glioma cells reduced cell growth and proliferation dramatically, resulting in an increased PTEN expression, which repressed PI3K/AKT pathway activation and prevented cancer development [33]. We mimicked the same mechanism in our model, which showed that induced SALL4 overexpression led to AKT phosphorylation at serine 473 residues and PTEN downregulation. The opposite effect was observed with SALL4 silencing in agreement with the former results. Otherwise, SALL4 overexpression positively regulated BCL2. Moreover, it has been described that SALL4 interacts with NuRD complex and regulates PTEN [21]. NuRD is one of four major types of ATP-dependent chromatin remodeling complexes, involved in various tumor progressions [34]. This complex comprises seven elements, including CHD3-4, HDAC1-2, RBBp4, MTA1 and MBD3 [22]. Each complex member functions in a cancer-type-dependent manner. Among them, RBBp4 is an essential subunit of the NuRD complex, which plays a key role during embryonic development. RBBp4 is also overexpressed in different cancer types, including TNBC, glioblastoma [35], hepatocellular carcinoma (HCC) and acute myeloid leukaemia [36]. The role of RBBp4 in HER2+ BC is still unclear. Here we first demonstrated that RBBp4 is significantly highly expressed in BC compared to healthy breast tissue. A deregulated SALL4–RBBp4/NuRD pathway results in tumor suppressors’ gene silencing, such as PTEN in HCC cells [23], leading to PI3K/AKT pathway activation. In our cancer model, we found that a high SALL4 expression increases the expression of NuRD complex units, including RBBp4. On the other hand, the opposite effect was observed in a trastuzumab-resistance cell line after SALL4 downregulation. Interestingly, PTEN direct regulation by SALL4 has been proposed, where co-expression of SALL4 with HDAC1 and/ or HDAC2 has been associated with low PTEN expression and poor prognosis in HCC [37]. Altogether, a secondary mechanism of PTEN regulation by SALL4 via RBBp4 (NuRD complex) and a physical interaction between SALL4 and RBBp4 can suppress PTEN expression, leading to PI3K/AKT pathway activation, finally increasing cell proliferation in HER2+ BC model.

Beforehand, it has been studied that SALL4 is involved in the EMT process in several types of cancer, including BC [38], gastric cancer [39], endometrial cancer [40], colorectal cancer [41] and esophageal squamous cell carcinoma [28]. Itou et al. suggested that SALL4 reduces the CDH1 expression to maintain cell dispersion (indicative of the migration ability) and promotes cell metastasis by activating focal adhesion [42], in a basal-like subtype of BC [43]. Moreover, it has been shown that SALL4 positively regulates EMT phenotype-associated proteins, such as ZEB1, Slug, Snail, and VIM and suppresses CDH1 in the TNBC and luminal subtype of BC [38]. We are the first to show that SALL4 also regulates EMT in the HER2+ BC subtype, with CDH1 being decreased and FN1, CTNNB1 and VIM expression enhanced by SALL4 overexpression. Moreover, it has been described that SALL4 positively regulates MYC by binding to the MYC promoter region [20]. MYC is an important TF that drives HER2+ tumors and EMT [44]. Overexpression of MYC in BC cells prompts the cells to be more proliferative and to acquire a more mesenchymal phenotype [44]. In our study, we found that the overexpression of SALL4 induced MYC expression, as well as EMT markers expression. Hence, in agreement with previous data, our results suggest that SALL4 induces the MYC expression as a result of induction of the EMT process, leading to a more aggressive cell phenotype.

Stemness and EMT processes go hand in hand. The poor survival rates reported in cancer are now thought to be due, among other causes, to cancer stem cells. It is well known that SALL4 is a regulator of cell stemness in biological development and tumor growth [45]. Thereby, we explored changes in stemness markers expression, including OCT4, SOX2 and NANOG, in SALL4 overexpressed cell lines. The results showed that high expression of SALL4 increases stemness markers. 

Taken together, we demonstrated that SALL4 expression is higher in HER2+ subtypes compared to other BC subtypes. Its expression is positively correlated with poor prognosis and aggressive properties of HER2+ BC. We identified that SALL4 induced high proliferation through the PI3K/AKT pathway and positively regulated EMT by targeting MYC. It suggests that SALL4 is holding back the trastuzumab effect partially by inducing a high proliferation rate. Further, we showed that SALL4 regulates the NuRD complex. The blocking SALL4 physical interaction with RBBp4 (a member of the NuRD complex) through a peptide has been proved in HCC [23] and can give an essential pharmacological approach to treating trastuzumab-acquired resistance in BC patients. In conclusion, our study provided a potential molecular mechanism related to SALL4-induced proliferation, EMT, and stemness in HER2+ BC cells. Further study of trastuzumab resistance models is needed to clarify the role of SALL4 in resistance. This mechanism may be a therapeutic target in the future to treat this specific BC subtype.

## 4. Materials and Methods

### 4.1. Cell Culture 

Human HER2+ BC BT474, SKBR3 cell lines were obtained from ATCC. BT474 is a luminal B/HER2+ cell line (ERB-B2 receptor tyrosine kinase 2 (ERBB2/HER2) amplified and overexpressed, cyclin D1 (CCND1) amplified, estrogen receptor (ER)+, progesterone receptor (PR)+, tumor protein p53 (TP53) and phosphatidylinositol-4,5-bisphosphate 3-kinase, catalytic subunit alpha (PI3KCA) mutated and wild type for breast cancer gene 1 (BCRA1) mutation) derived from adherent epithelial-like cells and initially collected from invasive ductal carcinoma of the breast tumor tissue. BT474R was obtained from BT474 cells cultured with appropriate medium supplemented with 15 μg/mL recombinant humanized HER2 monoclonal antibody trastuzumab (Herceptin, Genentech, South San Francisco, CA, USA) with increased doses until reach trastuzumab resistance. SKBR3 is an adherent epithelial HER2+ cell line (ERBB2 amplified and over-expressed, MYC amplified, CDH1 inactivated, ER-, PR-, TP53 mutated and wild type for BCRA1 mutation) derived from a metastatic site. Cell lines were grown in Dulbecco’s modified Eagle’s medium (DMEM) (Gibco, Carlsbad, CA, USA) supplemented with 10% fetal bovine serum (FBS; Gibco, Carlsbad, USA), 10,000 U/mL penicillin, 10,000 μg/mL streptomycin (Gibco, Carlsbad, USA) and 1% L-glutamine (200 mM) (100×) (Gibco, Carlsbad, USA). All the cells were cultured at 37 °C in a 5% CO_2_ atmosphere. The trastuzumab-resistant cell line BT474R was donated by F.R’ group from the Department of Pathology, IIS-Fundación Jiménez Díaz, Madrid, Spain. This cell line was cultured in the appropriate medium supplemented with 15 μg/mL recombinant humanized HER2 monoclonal antibody, trastuzumab (Herceptin, Genentech, USA). 

### 4.2. Plasmids and siRNA Transfection 

Cells were grown to 70–90% confluence and transfected using Lipofectamine^®^ 2000 re-agent (Invitrogen, Carlsbad, CA, USA), according to the manufacturer’s instructions. A volume of 1 μg/mL of pcDNA3.1 SALL4 (Addgene, Watertown, MA, USA) was used for SALL4 overexpression, whereas 1 μg/mL of pcDNA3.1 empty vector (Addgene, Watertown, USA) was used as transfection control. For the siRNA transfection, 100 nM SALL4 siRNA (si#1: s32816, si#2: s32817, Thermo Fisher Scientific, Waltham, MA, USA), as well as a negative control, was used in the BT474R cell line. Overexpression and knockdown efficiency was assessed at 72 h after transfection by RT-qPCR or western blot. Cells were incubated at 37 °C for 72 h, followed by cellular proteins and RNA extraction, and functional assays were performed.

### 4.3. RNA Extraction and Quantitative Real-Time PCR

Total RNA was extracted from the transfected cells by using TRIZOL reagent (Invitrogen, Carlsbad, CA, USA) according to the manufacturer’s instructions. High-Capacity cDNA Reverse Transcription kit (Applied Biosystems, Waltham, MA, USA) was used to synthesize 1000 ng of cDNA from total RNA. Following this, quantitative real-time PCR (RT-qPCR) was performed with a TaqMan^®^ 20× assay (Applied Biosystems, Waltham, USA) in a 384 reaction well plate (MicroAmp™ Optical 384-Well Reaction Plate with Barcode, Applied Biosystems™, Waltham, USA). Expression data were uniformly normalized to the internal control and relative gene expression was quantified using the 2^−ΔΔCt^ method. 

### 4.4. Cell Proliferation Assay

Cell proliferation was assessed using the WST-1 Assay Reagent Cell Proliferation (ab155902, Abcam, Cambridge, UK); 3 × 103 transfected cells were seeded in 96 well plates 72 h after transfection. Each day gap the media was replaced with fresh media supplemented with 15 μg/mL of trastuzumab. On day 7 of the experiment, the cell proliferation was measured using WST-1 reagent. A mix was prepared with 7% WST-1 reagent in phenol red-free media. A volume of 100 μL of the mix was added to the wells and incubated for 4 h at 37 °C. In a microplate reader, the absorbance was measured at 450–650 nm (background). 

### 4.5. Clinical Samples and RNA Isolation

Eighteen primary breast tumors were identified as HER2+ immunochemically by an expert pathologist and collected from the Clinical Oncology Department of the Hospital of Valencia. The patients were treated according to standard guidelines from 2004 to 2016. Patients underwent surgery after 6 months of neoadjuvant chemotherapy and anti-HER2 therapy including a combination of trastuzumab and pertuzumab. Adjuvant treatment was followed up to 1 year. Seven patients responded to treatment and 11 did not respond. Patients gave their written agreement to have their samples used in experiments. The project has ethical approval from the Hospital Clinico Research Ethics Committee and complies with all applicable ethical guidelines for research involving human subjects. Tumor samples were included in formalin-fixed paraffin-embedded tissues and recoverAll™ Total Nucleic Acid isolation kit (AM1975, Ambion, Waltham, MA, USA) was used to isolate total RNA from the tissue samples. The manufacturer protocol was followed for RNA isolation. An amount of 100 ng of total RNA was retro-transcribed using Reverse Transcription Kit (Applied Biosystems, Waltham, MA, USA), and 5 ng of cDNA was used for RT-qPCR for analyzing gene expression. The quantitative PCR analysis was performed as described earlier.

### 4.6. Western Blot and Co-Immunoprecipitation Analysis

Seventy-two hours after cell transfection, the protein was extracted from the whole lysate using RIPA Lysis buffer (89900, Thermo Scientific™, Waltham, MA, USA). The lysate was incubated for 30 min in ice and centrifuged for 30 min at 13,000 rpm at 4 °C. After collecting supernatant in a fresh tube, the protein concentration was determined using a BCA protein assay kit (23227, PierceTM BCA Protein Assay Kit, Thermo Scientific™, Waltham, USA). Proteins were separated on SDS PAGE and transferred to nitrocellulose membranes (1620115, Bio-Rad, Hercules, California, CA, USA). Following this, the membranes were blocked in 5% BSA for 1 h and then incubated with primary antibodies mentioned in Appendix A overnight at 4 °C. The following day, membranes were washed with Tris-buffer with 0.1% Tween^®^ 20 detergent (TBST) and subsequently incubated with the appropriate HRP-conjugated secondary antibodies for 1 hour at room temperature. Following this incubation, the membranes were washed and briefly incubated with Pierce™ ECL (32106, Thermo Scientific™, Waltham, USA) and the blot was developed using a chemiluminescence system (ImageQuant LAS400, GE Healthcare, Chicago, IL, USA). For co-immune precipitation, Protein A/G PLUS-Agarose beads were used (sc-2003, Santa Cruz Biotechnology, Dallas, TX, USA). We followed the manufacturer’s instructions and incubated the SALL4 antibody with 500 μg of cell lysate for 2 h at 4 °C, followed by the complex mixture being reacted with beads overnight at 4 °C on a rotating rack. The following day, the beads were washed three times with lysis buffer and separated on SDS PAGE. From there on, the same protocol followed as mentioned above for western blot and the blot was developed against RBBp4 protein using a chemiluminescence system. The appropriate control immunoglobulin G (IgG) was used for negative control. 

### 4.7. In Silico Survival Analysis

To evaluate the OS and DMFS linked to SALL4 mRNA expression in BC, a Kaplan–Meier plotter tool (KM-Plotter©) (http://kmplot.com/analysis/) was used, which was accessed on 5 February 2022 for figures in manuscript and for supplementary figures the KM plotter database was accessed on 12 October 2022. This tool uses a database containing different subtypes of BC Affymetrix microarray samples and associated gene expression with patients’ survival information. The median of the values included in the analysis was used as criteria to split the patients and the follow-up was 120 months. The hazard ratio (HR) with 95% confidence interval and log-rank *p*-value was calculated and presented. The results obtained were used to identify the prognostic value of SALL4 expressions on HER2+ BC.

### 4.8. Data Collection and Gene Expression Analysis from OncoDB and TCGA Database

The OncoDB public data set has been published recently in Nucleic acid research article [46]. The expression data of SALL4 and RBBp4 for breast invasive carcinoma (BRCA) were selected. The results were presented in a boxplot showing fold change and statistical significance comparing healthy with tumor tissue. On other hand, SALL4 and RBBp4 mRNA expressions were obtained from the TCGA database from different BC molecular subtypes.

### 4.9. Statistical Analysis

The student t-test was taken into consideration for comparing the sample and control groups. Standard deviation (SD) was included to measure the variations in all sets of data. The gene expression analysis from OncoDB and TCGA databases was performed by using the concept of false discovery rate (FDR)-adjusted *p*-value and log2 fold change. Statistical analysis was performed using the Shapiro–Wilk normality test and, based on the results of the normality test, parametric and non-parametric tests were applied to obtain the *p*-value. *p*-values < 0.05 were considered to be statistically significant.

## Figures and Tables

**Figure 1 ijms-23-13292-f001:**
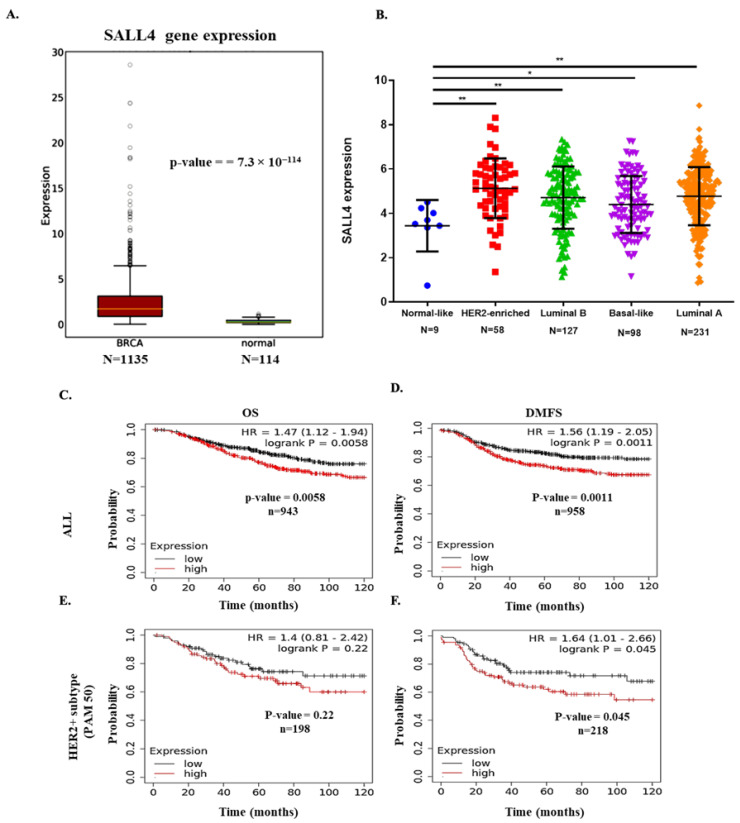
SALL4 expression analysis and prognostic value in breast cancer patients. (**A**) Boxplot to compare the RNA expression level of SALL4 in breast invasive carcinoma (BRCA) vs. normal breast tissue samples from the oncoDB database. (**B**) Gene expression analysis indicated the SALL4 expression level in the main intrinsic or molecular subtypes of breast cancer tissues from the TCGA database. Differential expression analysis is conducted with Student’s *t*-test, taking normal or low expression subtype as reference. (**C**–**F**) In-silico overall survival (OS) and distant metastasis-free survival (DMFS) analysis; (**C**,**D**) of SALL4 in all breast cancer patients (ALL) and (**E**,**F**) HER2+ subtypes patients from PAM50 database included in the Kaplan–Meier Plot tool. Patients were split by median value and the follow-up was 120 months. Student’s t-test compared the results. * *p*-value ≤ 0.05 and ** *p*-value ≤ 0.01.

**Figure 2 ijms-23-13292-f002:**
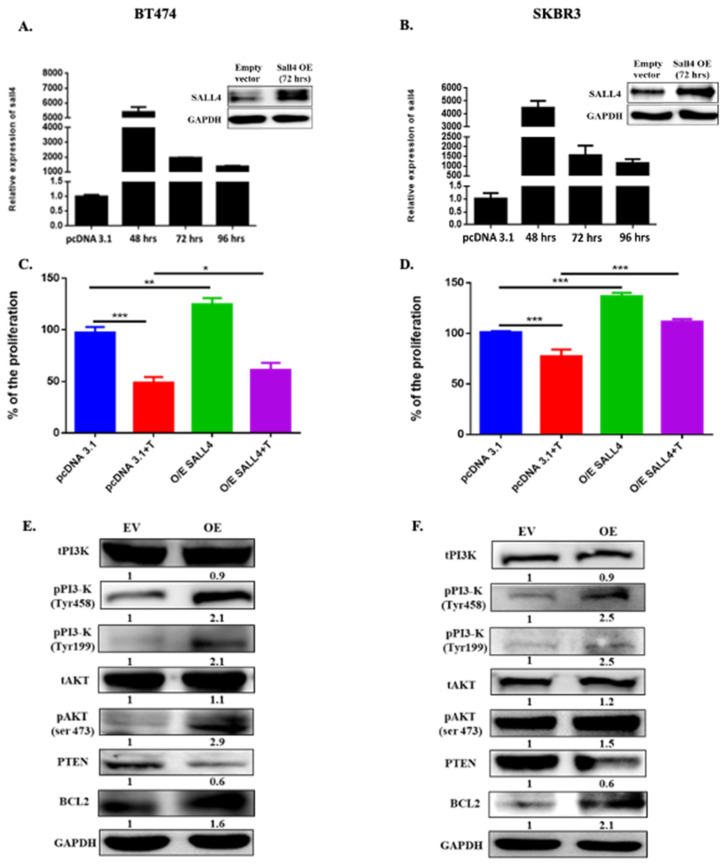
SALL4 induces proliferation by following PI3K/AKT pathway. (**A**,**B**) RT-qPCR and Western blot results show the overexpression efficiency of a plasmid (pcDNA3.1+/SALL4) for SALL4 in both HER2+ cell lines such as BT474 and SKBR3. (**C**,**D**) The proliferation of BT474 and SKBR3 cells with overexpression of SALL4 was measured by WST assay at 7 days. The graph shows the results of three independent experiments. Student’s t-test compared the results. * *p*-value ≤ 0.05, ** *p*-value ≤ 0.01 and *** *p*-value ≤ 0.001. (**E**,**F**) Protein expression levels were analyzed by Western blot following 72 h of transfection of plasmid for overexpression of SALL4. Expressions of PI3K/AKT pathway proteins, including PI3K, phospho-PI3 kinase (Tyr458 and Tyr199), phospho-Akt (Ser473), PTEN and BCL2 were shown. GAPDH expression was used as an internal control. The densitometry quantification was obtained by normalizing to GAPDH and for quantifying phosphorylation protein the data were obtained by taking the ratio of phosphorylated protein upon total protein. EV: Empty Vector: OE: Overexpression of SALL4.T: trastuzumab treatment.

**Figure 3 ijms-23-13292-f003:**
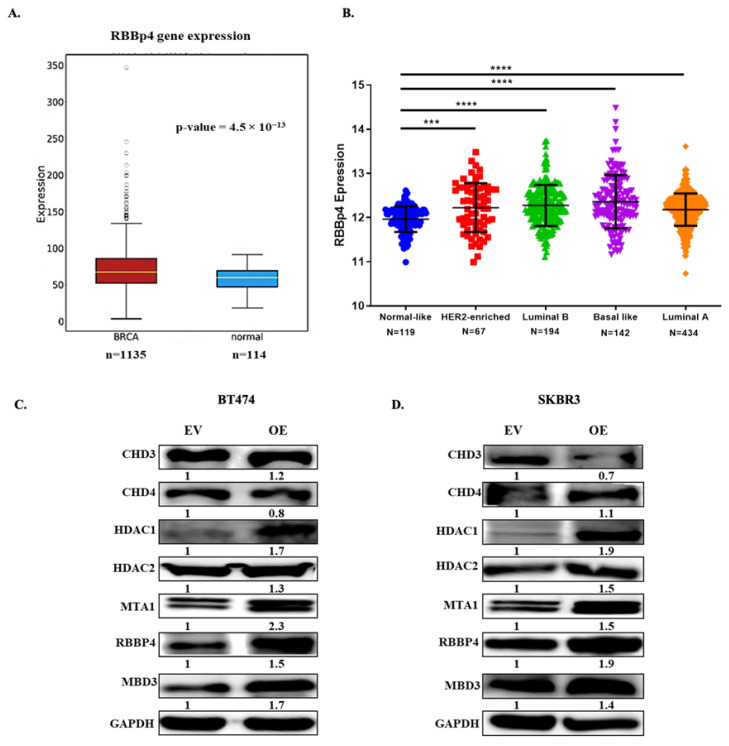
Expression of RBBp4 in public database and regulation of NURD complex through SALL4. (**A**) Boxplot to compare the RNA expression level of RBBp4 in breast invasive carcinoma vs. normal breast samples from the oncoDB database. (**B**) Gene expression analysis indicated the RBBp4 expression level in five main intrinsic or molecular subtypes of breast cancer tissues from the TCGA database. (**C**, **D**) Protein expression analysis of NURD complex members by upregulating SALL4 expression. Student’s t-test compared the results. *** *p*-value ≤ 0.001 and **** *p*-value ≤ 0.0001. EV: Empty Vector: OE: Overexpression of SALL4.

**Figure 4 ijms-23-13292-f004:**
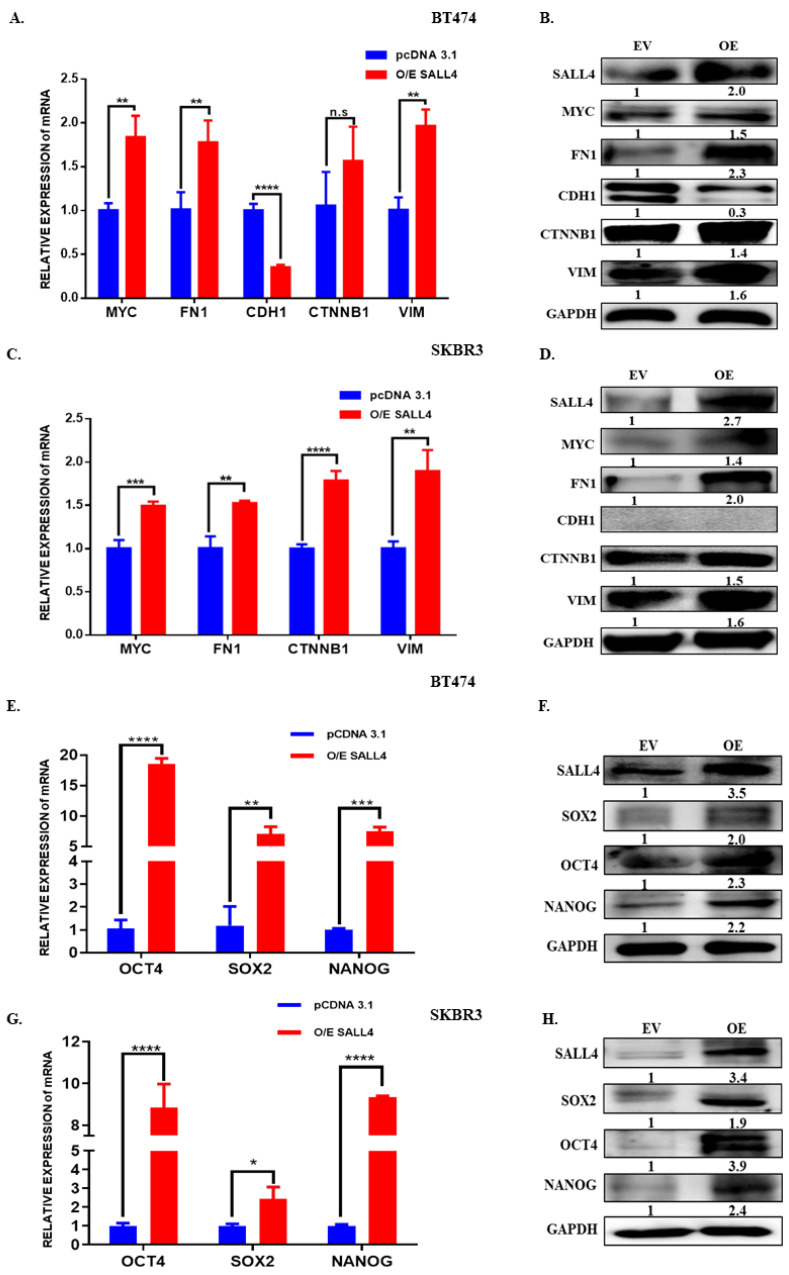
Modulation of EMT and stemness processes by SALL4 overexpression in HER2+ cells. (**A**–**D**) The expression of EMT-related markers and MYC was assessed by RT-qPCR and Western blot in both BT474 and SKBR3 cell lines with overexpression of SALL4. (**E**–**H**) mRNA and protein expression analysis of stemness markers by RT-qPCR and Western blot after upregulating SALL4 expression in BT474 and SKBR3. GAPDH expression was used as an internal control in both RT-qPCR and Western blot. Student’s t-test compared the results. * *p*-value ≤ 0.05 and ** *p*-value ≤ 0.01, *** *p*-value ≤ 0.001 and **** *p*-value ≤ 0.0001. EV: Empty Vector: OE: Overexpression of SALL4. n.s: not significant.

## Data Availability

Not applicable.

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
