# Peer review of "Role of SALL4 in HER2+ Breast Cancer Progression: Regulating PI3K/AKT Pathway"

_ijms, 2022, doi:10.3390/ijms232113292_

Round 1

Reviewer 1 Report

This is an interesting  manuscript for people working on breast cancer biology.

The manuscript is well written and scientific sounding. However some minor points needs to be adressed.

1-More complete legends must be provided.

2- Figure 2F are not clear, a better resolution should be provided and WB results quantified.

3- Cell lines should be better characterized in Methodology.

4- There are minor spelling mistakes in the text. 

Author Response

We are grateful for the comments of the reviewers that have eminently improved our work. We have responded to one by one the questions.

Reviewer 1

  1. More complete legends must be provided.

Thank you for the valuable suggestions. We have completed the figure legends with additional information.

  1. Figure 2F is not clear, a better resolution should be provided and WB results quantified.

We agree with you about the upgradeable quality of western blot. We have increased the resolution of the western blot as much as possible. We have also quantified the western blot bands. Numerical quantification data have been added below each western blot figure and quantification graphs have been incorporated in supplementary material together with the original western blots. The AKT phosphorylation level has been calculated by the ratio of pAKT/tAKT after normalizing to GAPDH (endogenous control).

  1. Cell lines should be better characterized in Methodology

Additional cell lines characteristics have been included in the methodology section.

  1. There are minor spelling mistakes in the text

The manuscript has been viewed again in extensive way for checking spelling mistake and grammars.

Reviewer 2 Report

The manuscript by Pattanayak et al titled “Role of SALL4 in HER2+ breast cancer progression: regulating PI3K/AKT pathway” provides evidence in the role of SALL4 in promoting the proliferation and potentially drug resistance in HER2+ breast cancer cells. SALL4 is a well-known biomarker and target of multiple types of cancer, and it has been known to promote proliferation, EMT and drug resistance in cancer types including breast. The authors expand our current knowledge of SALL4 in breast cancer by focusing on the HER2+ subtype. They used both trastuzumab-sensitive and -resistant cell lines to show that overexpressing SALL4 enhances proliferation (likely via the PI3K/AKT pathway). The data presented in the manuscript are mostly clear, however some of the conclusions lack experimental support. Below are detailed comments and questions:

1. Throughout the manuscript, the authors use SALL4 overexpression in trastuzumab-sensitive cell lines and SALL4 knockdown (KD) in resistant cell lines to determine its role in proliferation and etc. It would strengthen the conclusions if KD experiments were also performed on these trastuzumab-sensitive cell lines in parallel to overexpression.

2. To better compare the sensitivity of cells to trastuzumab (T) in the presence vs. absence of SALL4 overexpression (Fig 2C and D), one should calculate the ratio of proliferation w/ T to that w/o T (i.e. red/blue vs purple/green instead of red vs. purple). I wonder if the conclusion “cells with SALL4 overexpression decreased their trastuzumab response compared to cells with basal SALL4 levels” still holds true.

3. It would take a lot more to support a conclusion like “these results suggested that SALL4 regulates HER2+ pathway through PI3K/AKT pathway, which leads to cell growth and tumor proliferation, favoring trastuzumab resistance, and may play a significant role in BC progression.” Can the authors show with experimental results that blocking PI3K pathway abolishes SALL4’s effect on proliferation and trastuzumab resistance?

4. The authors claim that “Taken together, these data suggested that the interaction between SALL4 and RBBp4 (NuRD complex) inhibits PTEN expression and activates the PI3K/AKT pathway that helps cells to survive and to resist therapy” when there is nothing to demonstrate interaction between SALL4 and RBBp4 in the tested cell lines. Alternatively, one could use a mutant of SALL4 that does not interact with RBBp4 (if available) to demonstrate this point.

5. It is unclear what high and low expression of SALL4 mean in Fig 1C-F. What’s the criterion for stratifying patients into these two categories?

6. The statement “the results showed that in both cases more expression of SALL4 was related to less OS (p-value = 0.0058 for ALL and non-significant for HER2+) (Fig. 1C and 1E)” does not hold when the HER2+ result is not significant.

7. Were western blots done in biological replicates (e.g. Fig2E&F, 3C&D and etc)? Some of the expression phenotypes are very subtle, so quantifications and statistical tests would be required to draw convincing conclusion.

Author Response

Reviewer 2

  1. Throughout the manuscript, the authors use SALL4 overexpression in trastuzumab-sensitive cell lines and SALL4 knockdown (KD) in resistant cell lines to determine its role in proliferation and etc. It would strengthen the conclusions if KD experiments were also performed on these trastuzumab-sensitive cell lines in parallel to overexpression.

We observed high expression of SALL4 in the resistant cell line and low expression of SALL4 in the sensitive/parental cell lines. So, for that reason, we did not seek to eliminate SALL4 expression in the parental cells, where there is already a low basal level. For further clarification, we have included the figures shown below (Figure 1A-B) as Supplementary Figure 2A and 2B, as well as in the text of the manuscript.

  1. To better compare the sensitivity of cells to trastuzumab (T) in the presence vs. absence of SALL4 overexpression (Fig 2C and D), one should calculate the ratio of proliferation w/ T to that w/o T (i.e. red/blue vs purple/green instead of red vs. purple). I wonder if the conclusion “cells with SALL4 overexpression decreased their trastuzumab response compared to cells with basal SALL4 levels” still holds true.

We thank the reviewer for the observation. Truly, the sentence get place to confusion because we want to indicate that to have high SALL4 expression level give the cell line advantages against the trastuzumab treatment, in the sense that SALL4 conduce to increase proliferation to contraries the effect of the treatment. In other words, trastuzumab is less effective but not because a resistance mechanism, at least no proved yet, but a more proliferative behaviour of the cells. Indeed, if we compare the proliferation decreased ratio before and after trastuzumab treatment between cells with or without SALL4 overexpression, the difference is not significative, despite the trastuzumab proliferation reduction is less when SALL4 overexpressed. We have changed the sentence to clarify this concept.

  1. It would take a lot more to support a conclusion like “these results suggested that SALL4 regulates HER2+ pathway through PI3K/AKT pathway, which leads to cell growth and tumor proliferation, favoring trastuzumab resistance, and may play a significant role in BC progression.” Can the authors show with experimental results that blocking PI3K pathway abolishes SALL4’s effect on proliferation and trastuzumab resistance?

This is a great point to notice. We completely agree that, it would support strongly our conclusion. Unfortunately, in short time periods it is difficult to use the inhibitor against a signalling pathway and conduct the all-western experiments.

Nevertheless, several evidences relate SALL4 to PI3K/AKT pathway. SALL4 has a binding site at the promoter regions of PTEN and SALL1 which co-occupied by NuRD components, suggesting that SALL4 represses the transcriptions of PTEN and SALL1 through its interactions with the Mi-2/NuRD complex1. We have also evaluated the SALL4 expression in the PTEN mutated MDA-MB-468 BC cell line (see below, Fig. 1C) 2. The result shown high basal level of SALL4 suggesting that SALL4 is more expressed in absent/mutation of PTEN. When we have performed loss and gain functions of SALL4 in the HER2+ cell lines (both in parental and acquired trastuzumab resistance) we have found an inverse regulation of PTEN expression (Fig 2E and F, Supplementary Fig 2E). PTEN has been extensively proved as a master regulator of PI3K/AKT pathway and this pathway has been implicated in trastuzumab resistance in HER2-overexpressing BC3,4,5,6.

Additionally, it has been reported that SALL4 physically interacts with retinoblastoma 4-binding protein (RBBp4), which regulated via methylation of the PTEN promoter or via histone deacetylase-containing complexes NuRD/Mi27. We have confirmed the SALL4 and RBBp4 interaction in our model by co-immune precipitation (see below, Fig. 1D). We have also found regulation of NuRD complex units by SALL4 loss and gain function. Finally, the experiments of Liu BH et al. with a peptide that avoid the physical contact between SALL4 and RBBp4 show PTEN activation and negatively regulation the PI3K/AKT pathway 8.

  1. The authors claim that “Taken together, these data suggested that the interaction between SALL4 and RBBp4 (NuRD complex) inhibits PTEN expression and activates the PI3K/AKT pathway that helps cells to survive and to resist therapy” when there is nothing to demonstrate interaction between SALL4 and RBBp4 in the tested cell lines. Alternatively, one could use a mutant of SALL4 that does not interact with RBBp4 (if available) to demonstrate this point.

To prove the interaction between SALL4 and RBBp4 in our model, we have done a co-immunoprecipitation experiment of SALL4 and RBBP4, which gave us a physical interaction between these two proteins (see below, Fig1D). We have also included this data into our manuscript as supplementary Figure 3A and also in text.

  1. It is unclear what high and low expression of SALL4 means in Fig 1C-F. What’s the criterion for stratifying patients into these two categories?

To stratify the patients, we have used the median value of the data included in the analysis. The reason for using this criterion is because the median is the value of the central position of a variable in an ordered data set.

Information about the criteria used has been added in materials and methods.

  1. The statement “the results showed that in both cases more expression of SALL4 was related to less OS (p-value = 0.0058 for ALL and non-significant for HER2+) (Fig. 1C and 1E)” does not hold when the HER2+ result is not significant.

Thanks for the comment. The reviewer is right, since with the figure shown (not significant) the relationship with survival cannot be confirmed, although a trend can be seen. However, the analysis shown was performed using the HER2+ data included in the PAM5O database of the Kaplan Meier Plotter tool. When we analyze the relationship between the expression of SALL4 and survival using HER2+ patients from the HER2 array and the StGallen databases, in both cases the relationship with survival is clear, associating higher expression of SALL4 with worse survival both in the case of OS and of DMFS.

In order to clarify the results, we have included in the text that the data obtained for HER2+ vary according to the database analyzed and suggest that the relationship between SALL4 and survival is the same as that shown in the analysis of ALL breast cancer patients.

The new graphs obtained from the analysis of the HER2 array and the StGallen databases have been included in supplementary material (supplementary Figure 1B).

  1. Were western blots done in biological replicates (e.g. Fig2E&F, 3C&D and etc)? Some of the expression phenotypes are very subtle, so quantifications and statistical tests would be required to draw convincing conclusion.

Western blots have been quantified and values have been included below the bands. Quantification plots have been added in supplementary material alongside the original western blots.

MDA-MB-468

c

D

Figure 1.  (A) Relative expression of SALL4 was determined in BT474 HER2+ BC cell line parental and trastuzumab acquired resistance BT474R HER2+ BC cell line.  (B) SALL4 protein expression was determined by western blot in mentioned cell lines. Student's t-test was used to analyse the significant differences. ***P ≤ 0.001. (C) SALL4 protein expression was determined by western blot in MDA-MB-468 cell line. (D) Co-Immunoprecipitation western blot revealing the physical interaction between SALL4 and RBBp4 in SKBR3 and BT474 parental HER2+BC cell lines and BT474R trastuzumab acquired resistance HER2+BC cell line.

References

  1. Lu J, Jeong H, Kong N, et al. Stem Cell Factor SALL4 Represses the Transcriptions of PTEN and SALL1 through an Epigenetic Repressor Complex. PLoS One. 2009;4(5):1-13. doi:10.1371/journal.pone.0005577
  2. Marty B, Maire V, Gravier E, et al. Research article Frequent PTEN genomic alterations and activated phosphatidylinositol 3-kinase pathway in basal-like breast cancer cells. 10(6):1-15. doi:10.1186/bcr2204
  3. Paplomata E, Regan RO. The PI3K / AKT / mTOR pathway in breast cancer : targets , trials and biomarkers. Ther Adv Med Oncol Rev. 2014:154-166. doi:10.1177/1758834014530023
  4. Ritter CA, Perez-Torres M, Rinehart C, et al. Human breast cancer cells selected for resistance to trastuzumab in vivo overexpress epidermal growth factor receptor and ErbB ligands and remain dependent on the ErbB receptor network. Clin Cancer Res. 2007;13(16):4909-4919. doi:10.1158/1078-0432.CCR-07-0701
  5. Fiszman GL, Jasnis MA. Molecular Mechanisms of Trastuzumab Resistance in HER2 Overexpressing Breast Cancer. Int J Breast Cancer. 2011;2011:1-11. doi:10.4061/2011/352182
  6. Kumar A, Xu J, Brady S, et al. Tissue transglutaminase promotes drug resistance and invasion by inducing mesenchymal transition in mammary epithelial cells. PLoS One. 2010;5(10). doi:10.1371/journal.pone.0013390
  7. Mcloughlin NM, Mueller C, Grossmann TN. Perspective The Therapeutic Potential of PTEN Modulation : Targeting Strategies from Gene to Protein. Cell Chem Biol. 2018;25(1):19-29. doi:10.1016/j.chembiol.2017.10.009
  8. Liu BH, Jobichen C, Chia CSB, et al. Targeting cancer addiction for SALL4 by shifting its transcriptome with a pharmacologic peptide. Proc Natl Acad Sci U S A. 2018;115(30):E7119-E7128. doi:10.1073/pnas.1801253115

Round 2

Reviewer 2 Report

Authors have addressed my comments